# Genetic Susceptibility to Endometrial Cancer: Risk Factors and Clinical Management

**DOI:** 10.3390/cancers12092407

**Published:** 2020-08-25

**Authors:** Thilo Dörk, Peter Hillemanns, Clemens Tempfer, Julius Breu, Markus C. Fleisch

**Affiliations:** 1Department of Gynecology and Obstetrics, Comprehensive Cancer Center, Hannover Medical School, 30625 Hannover, Germany; hillemanns.peter@mh-hannover.de; 2Department of Gynaecology, Marien-Hospital, Ruhr University of Bochum, 44625 Herne, Germany; Clemens.Tempfer@elisabethgruppe.de; 3Department of Gynecology and Obstetrics, University of Witten/Herdecke, 42283 Wuppertal, Germany; Julius.Breu@helios-gesundheit.de (J.B.); Markus.Fleisch@helios-gesundheit.de (M.C.F.)

**Keywords:** endometrium, carcinoma, genetics, Lynch syndrome, Cowden syndrome, targeted therapy

## Abstract

Endometrial cancer (EC) is the most common cancer affecting the female reproductive organs in higher-income states. Apart from reproductive factors and excess weight, genetic predisposition is increasingly recognized as a major factor in endometrial cancer risk. Endometrial cancer is genetically heterogeneous: while a subgroup of patients belongs to cancer predisposition syndromes (most notably the Lynch Syndrome) with high to intermediate lifetime risks, there are also several common genomic polymorphisms contributing to the spectrum of germline predispositions. Germline variants and somatic events may act in concert to modulate the molecular evolution of the tumor, where mismatch-repair deficiency is common in endometrioid endometrial tumors whereas homologous recombinational repair deficiency has been described for non-endometrioid endometrial tumors. In this review, we will survey the currently known genomic predispositions for endometrial cancer and discuss their relevance for clinical management in terms of counseling, screening and novel treatments.

## 1. Risk Factors, Biology and Genetics

### 1.1. Epidemiology

Endometrial cancer (EC) develops in the inner layer of the uterus from the glandular epithelial sheet that covers the luminal surface and secretes substances essential for normal menstruation and embryonic development. Endometrial cancer is the most common cancer affecting the female reproductive organs in higher-income states [1,2,3]. Women in the US have a 2.6% lifetime risk of developing endometrial cancer, with some racial disparity documented [4]. It is commonly encountered in postmenopausal patients, although some 25% of cases occur prior to menopause, with some 5% in patients younger than 40 years old [1,5]. Most women diagnosed with endometrial cancer (EC) have well-differentiated cancers with endometrioid histology associated with early-stage disease and favorable outcomes, however there are clinically aggressive histologic subtypes of the disease, such as the serous histotype [2]. Five year overall survival ranges from 74% to 91% in patients without metastatic disease [3].

Endometrial cancer is the most strongly hormone- and excess weight-related cancer [1,6,7]. Both factors are correlated through the elevated estrogen levels associated with greater body weight in older women, since after menopause, androgens are converted to estrogens through the enzyme aromatase, found in adipose cells. Other risk factors, including reproductive factors such as parity, may also reflect the influence of sex steroid hormones. The endometrium is a dynamic tissue that undergoes frequent remodeling in response to oscillating levels of estrogen and progesterone, implying a recurrent need for clonal expansion and repopulation from progenitor cells [8,9].

However, risk factors in patients with uterine serous carcinoma seem to differ from those in women with endometrioid carcinoma, suggesting that there may be at least two different pathways of endometrial carcinogenesis. While a strong association was observed for high body mass index (BMI) in the endometrioid carcinoma cases, serous carcinomas were not strongly associated with BMI [10]. Parity, oral contraceptive use, cigarette smoking, age at menarche and diabetes were associated with both histological groups to similar extents [10,11].

### 1.2. Biology

The genomic landscapes accompanying clonal proliferation and malignant transformation of endometrial cells have been studied through next-generation sequencing in endometrial cancer as well as in normal endometrial epithelium [12,13,14,15,16]. Clonal expansion and a basal mutational burden can already be observed in single glands of normal endometrium, which accumulate somatic variants by age at an estimated rate of 29 variants per gland per year [16]. The basal mutational rate is about five times higher in endometrial cancer cells, which additionally can adopt a mutator phenotype with highly increased levels of specific mutational signatures indicating defective repair of mismatches, oxidative damage or breaks in their genomic DNA [12,13,14,16]. Large-scale molecular analysis of endometrial tumors grouped endometrial cancers into four molecular classifications: POLE-ultramutated, microsatellite instability-hypermutated, copy-number low (also called endometrioid-like) and copy-number high (also called serous-like) [12]. There is marked correlation between molecular and histological data: endometrioid histology is mostly represented in the first three clusters, which are distinguished by different degrees of microsatellite instability, DNA methylation and copy-number alterations, whereas the copy-number-high cluster covers most of the serous histology and tends to be a high grade and stage, with a high rate of recurrence and poor disease related outcome [12]. A comparison of frequently mutated genes revealed that activating variants in a group of known cancer driver genes, such as *PIK3CA*, *PIK3R1*, *ERBB2* or *KRAS*, can already be found in separate glands of normal epithelium and may occur as early as in the first two decades of life [16]. On the other hand, cancer driver genes known to be commonly mutated in endometrial cancer such as *TP53*, *PTEN* or *CTNNB1* were rarely mutated in the normal endometrium. This suggests a model in which single endometrial glands may first acquire driver mutations in genes that aid in tissue homeostasis but have limited capacity of malignant transformation. Endometrial cancer may then develop from these cells after additional mutational or epigenetic events impairing additional tumor suppressor or caretaker genes [16,17]. In this model, both stages would be driven by age-dependent accumulation of somatic variants and the neoplastic evolution started in early lifetime can take several decades, but the development would be accelerated if there is inherited variation in any of the cancer driver genes or in genes impacting on mutational repair, resulting in a genetic predisposition. Pathological examinations indicate that the first very subtle tissue alterations in a genetically predisposed woman can be seen in endometrial glands where cells ultimately lose the expression of the affected mismatch repair gene [18].

### 1.3. Genetic Predisposition

#### 1.3.1. Spectrum of Hereditary Factors

A family history of endometrial cancer is associated with a two-to-threefold increased risk of endometrial cancer [19,20]. Although some of the associations between family history and endometrial cancer risk may be attributable to shared environmental or lifestyle risk factors, twin studies have estimated heritability between 27% and 52% [20,21,22,23]. Furthermore, colorectal, breast and stomach cancers co-occur with endometrial cancer at significantly higher frequencies in patient compared to control families [20]. A genetic predisposition for both colorectal and endometrial cancer was first observed in women with Lynch Syndrome, a hereditary cancer syndrome that accounts for 2–3% of all endometrial cancer cases [24,25]. However, family history is still associated with a risk of endometrial cancer after excluding Lynch Syndrome patients, indicating that additional genetic risk factors exist for familial endometrial cancer [20]. More recent work has uncovered further endometrial cancer susceptibility candidate genes [26] as well as several genomic loci harboring common low-risk variants [27] (Figure 1, Table 1); the current status of this ongoing research will be briefly reviewed below, followed by a survey and discussion of the clinical implications.

#### 1.3.2. Lynch Syndrome

Lynch Syndrome (LS) represents one of the most common hereditary cancer syndromes [24], occurring at an incidence of 1/100–1/180 [28]. Due to its involvement of colorectal carcinomas, it is also known as Hereditary Non-Polyposis Colorectal Cancer (HNPCC). However, already the first LS pedigrees described had shown a high incidence of cancers within the uterus [29,30], and it is now well established that endometrial cancer is a frequent part of the cancer spectrum in LS [20,31,32]. Endometrial tumors from LS patients are often poorly differentiated, tend to have tumor-infiltrating lymphocytes and tend to involve the low uterine segment [4]. By contrast with patients who have sporadic endometrioid low-grade tumors, LS patients do not tend to have a particularly high body mass index [33].

LS is caused by germline pathogenic variants affecting one of four genes encoding the DNA mismatch repair (MMR) system components: *MLH1*, *MSH2*, *MSH6* and *PMS2*. These genes encode proteins that are required for the repair of mismatches in the base-pairing of genomic DNA, thereby ensuring the integrity of genomic information [34]. MMR deficiency results in a hypermutability with 100–1000 fold increased mutational rates [35]. Due to slippage in the replication process, runs of repetitive nucleotides are prone to base-pairing errors, resulting in the diagnostic observation of “microsatellite instability” in MMR-deficient tumors. As MMR proteins also signal through the ATM/ATR pathway, MMR-deficient cells fail to undergo damage-induced cell cycle arrest and apoptosis [28,36,37,38].

Individuals with LS are usually heterozygous for a pathogenic MMR gene variant, and somatic inactivation of the second allele (“second hit”) is required to develop a carcinoma from the respective cell type. This commonly occurs via a loss of heterozygosity, e.g., through deletion of the second allele. Another mode of somatic inactivation is gene promoter methylation, which is the main cause of sporadic tumors with MMR deficiency, particularly for *MLH1* [39,40]. In a small proportion of patients, the hypermethylation can be induced by constitutional deletions of the neighboring gene *EPCAM* (adjacent to *MSH2*); these germline deletions thus represent additional hereditary factors [41,42]. Constitutional hypermethylation has also been reported for *MHL1*, with some cases occurring de novo in non-transmissible Lynch Syndrome, but in other cases the underlying epimutations can be inherited [41,43,44]. In addition, few reports exist about deletions or inversions affecting both *MLH1* and *LRRFIP2* (upstream of *MLH1*) that either create expressed, but non-functional, gene fusions or inactivation of both genes [45,46,47].

Rare cases with biallelic hereditary variants in MMR genes and a constitutional MMR deficiency (CMMRD) have also been described, including few endometrial cancer patients [48,49,50]. The syndrome is also known as Turcot Syndrome. In such patients, where the “second hit” is already inherited, hematological or brain tumors in addition to LS-associated carcinomas already manifest at childhood and endometrial cancer can occur in young females, suggesting intensified gynecological surveillance starting at age 20 [49].

Data from the Prospective Lynch Syndrome Database (PLSD) indicate that the average lifetime risks of LS-associated endometrial cancers in heterozygotes are high. Risks to age 75 years for heterozygous *MLH1* variant carriers were 37% (30–47%), for *MSH2* variant carriers 49% (40–61%), for *MSH6* variant carriers 41% (29–62%) and for *PMS2* variant carriers 13% (5–50%) [32]. These risk estimates are in a similar range as for colorectal cancer and are 3–4 times higher than for ovarian cancer [24,32]. *PMS2* variants seem to confer a somewhat lower endometrial cancer risk than variants in the other three genes [32,51]. Perhaps consistent with a more moderate penetrance, it has turned out that *PMS2* variants may be more common in colon cancer patients with a less pronounced family history that would not fulfill the Amsterdam criteria for LS syndrome [52]. It remains to be seen whether this holds true for endometrial cancer.

MMR deficiency in endometrial cancer may be best assessed using immunohistochemical (IHC) staining for the four major DNA mismatch repair proteins (MLH1, MSH2, MSH6 and PMS2), which appears to outperform the classical microsatellite instability testing [53]. Loss or abnormality of nuclear staining for any of the proteins would indicate MMR deficiency. This is a common observation also in sporadic endometrial tumors of the endometrioid type [54], and genetic testing at the germline level will then be required to distinguish between LS and sporadic origin. As will be discussed later, this will make an important difference in the medical counseling and surveillance of these patients.

#### 1.3.3. Cowden Syndrome

Cowden Syndrome (CS), named after the first described patient Rachel Cowden, is an autosomal dominant syndrome that has been associated with a germline mutation in *PTEN* [55,56,57] (for review see [58]). This gene encodes a phosphatase involved in cell signaling pathways affecting cell proliferation and survival. CS is a hamartoma and cancer syndrome that affects about 1 in 200,000 individuals and predisposes to an increased lifetime risk for cancers of the breast, thyroid, kidney, endometrium, colon and for melanoma [59]. Endometrial cancer occurs in 21–28% of CS patients [60,61,62] and can develop at a very early age [63,64]. Immunohistochemistry shows loss of PTEN staining in endometrial cancers of CS patients, however, somatic mutation of *PTEN* and therefore loss of PTEN expression are also common observations in sporadic endometrial cancer [65,66]. Germline pathogenic *PTEN* variants are very rare in endometrial cancer patients outside of CS families [67].

CS is genetically heterogeneous, however, and at less stringent criteria only some 25% of CS patients harbored germline *PTEN* loss-of-function variants [68]. About 10% of Cowden and Cowden-like syndrome individuals without a detected *PTEN* variant have been found to carry germline gain-of-function variants in *AKT1* or *PIK3CA* that are predicted to act in the same PI3K/AKT signaling pathway, which is inhibited by PTEN [68]. The relative proportion of endometrial cancers in this group of CS patients remains to be established.

#### 1.3.4. Further Candidate Genes

##### *POLE* and *POLD1*

Germline variants within the exonuclease domains of the DNA replication and proof-reading polymerases POLD1 and POLE have been identified as dominantly inherited predispositions in colorectal cancer and have also been implicated in susceptibility to endometrial cancer [69,70,71,72,73]. At the presently low numbers, the relative risks for endometrial cancer have not yet been quantified but pedigrees were consistent with a hereditary predisposition. Biallelic *POLE* variants have also been found to cause a growth retardation and immunodeficiency syndrome [74,75]; however, the loss-of-function variants causing this syndrome seem to be distinct from the missense variants in the catalytic domain that confer the non-syndromic susceptibility to cancer [74].

##### *MUTYH* 

The *MUTYH* gene encodes a protein of the base excision repair system, which repairs oxidative DNA damage. MUTYH-associated polyposis is an autosomal recessively inherited predisposition to adenomatous polyposis and colorectal cancer. While biallelic variants have been associated with a 75% lifetime risk of colon cancer, monoallelic variants may confer an only 7% lifetime risk of colon cancer [76]. Some results have been presented that heterozygotes for *MUTYH* variants may also carry an about two-fold increased risk for endometrial cancer [77,78]. However, the evidence is limited so far and larger replication studies are warranted. A splice variant of *MUTYH*, c.934-2A>G, occurs at a carrier frequency of up to 3% in East Asian populations but has not firmly been linked to disease.

##### *NTHL1* 

The *NTLH1* gene encodes another protein of the base excision repair system in which mutations were reported as an autosomal recessively inherited predisposition to adenomatous polyposis and colorectal cancer [79]. Homozygosity for a germline nonsense variant in *NTLH1* was identified in multiple polyposis-affected patients from three unrelated families, all including women who developed an endometrial malignancy [79]. A subsequent study of another 17 families with biallelic *NTLH1* variants indicated a high incidence of polyposis coli and breast carcinomas but also identified endometrial cancer patients in 4 of the 17 families [80]. According to these findings, constitutional *NTHL1* deficiency underlies a high-risk hereditary multitumor syndrome that appears to predispose homozygotes to colon cancer, breast cancer and endometrial cancer. It is unknown whether *NTHL1* monoallelic variants also confer some increase in cancer risk, as has been suggested for *MUTYH* heterozygotes, but from the available evidence the heterozygote risk, if any, is likely to be small.

##### *BRCA1* 

A multinational cohort study of the Breast Cancer Linkage Consortium involving 11,847 *BRCA1* variant carriers reported a significant two-to-three-fold increase in the risk of endometrial cancer [81]. However, the interpretation of these results is complicated by the fact that *BRCA1* variant carriers may take tamoxifen, which is known to increase endometrial cancer risk (for a more extensive review see [26]). Subsequent studies identified a larger risk for those women taking tamoxifen compared to those without, and conflicting results have been obtained whether there is any tamoxifen-independent residual risk of endometrial cancer for carriers of a pathogenic *BRCA1* or *BRCA2* variant [82,83]. Evidence has been presented that *BRCA1* variant carriers after risk-reducing salpingo-oophorectomy may have an increased risk of the more aggressive but rarer serous/serous-like endometrial cancers, though no significant risk of the endometrioid subtype cancer [84], and that these *BRCA1*-associated endometrial cancers are associated with an unfavorable outcome [85]. However, more research is required to corroborate these observations.

#### 1.3.5. Low-Penetrance Susceptibility Loci

A growing body of evidence has linked single-nucleotide variants with polymorphic frequencies at various genomic regions to an altered risk of endometrial carcinoma. This has mainly been accomplished through genome-wide case-control association studies (GWAS) conducted by the Endometrial Cancer Association Consortium [27]. A first genome-wide significant endometrial cancer susceptibility region was identified in the 5′-portion of the gene *HNF1B* [86]. Subsequent GWAS with higher coverage and larger sample size yielded additional 15 genomic risk loci so far [27,87,88,89], including some known pan-cancer gene loci such as *MYC*, *CDKN2A* or *NF1*. Interestingly, a comparison with GWAS results from other clinical entities revealed an overlap of six loci with traits associated with steroid hormone levels [27]. Furthermore, credible causal endometrial cancer risk variants were enriched at epigenetic marks that were activated by estrogen stimulation in endometrial cancer cells, supporting the causal role of estrogen in this cancer (see O’Mara et al., 2019 [27]). One initially genome-wide significant hit was the genomic locus for *AKT1*, a gene also implicated in Cowden Syndrome (see Section 1.3.3). Subsequent functional studies showed that the most likely causal variant in this region, the risk allele of rs2494737, generates a binding site for the transcription factor YY1 that would stimulate *AKT1* expression [90]. However, with increasing sample size the *AKT1* locus has fallen below the genome-wide significance threshold, and further studies will be needed to finally prove its relevance.

Although each of the 16 identified genome-wide significant variants has only a small effect on risk, they are common polymorphisms with a cumulative contribution to the familial relative risk of the disease. It has been estimated that common genetic variants of the type that can be tagged by standard GWAS arrays potentially account for approximately 28% of the familial relative risk of endometrial cancer, and that the 16 risk variants identified to date account for approximately one quarter of this figure, suggesting that many more genetic risk variants remain to be found [27,89]. Meta-analyses are presently being performed to identify additional variants relevant for endometrial cancer and a related disease. For instance, a cross-cancer GWAS for endometrial and colon cancer identified a risk variant in *SH2B3* [91], and a cross-disease GWAS for endometriosis and endometrial cancer identified a risk region within the *PTPRD* gene [92]. It is likely that more such meta-analyses will uncover additional loci.

With increasing number of low-penetrance variants for endometrial cancer, it may be possible to construct polygenic risk scores that should be helpful in future genetic risk prediction and counseling [93]. Equally importantly, every newly identified region adds to our knowledge about the etiology of the disease and can point to possible pharmacological targets. In addition, single nucleotide polymorphism genotypes have already proven very helpful in corroborating epidemiological and clinical observations by detecting potentially causal relationships between physiological traits and endometrial cancer through Mendelian randomization analyses [27,94,95,96,97].

## 2. Clinical Implications and Management

### 2.1. Screening for Hereditary Syndromes among Patients with Endometrial Cancer

In order to identify recently published relevant clinical studies in the field we performed a PubMed search using a strategy as presented in Figure 2 and checked cross references.

Patients with hereditary forms of endometrial cancer are mostly diagnosed at a younger mean age (48 years) compared to patients in an unscreened population (68 years). About 10% of all patients are diagnosed prior to age 50. Moreover, in female Lynch Syndrome mutation carriers, endometrial cancer often precedes other cancers and therefore can be considered a “sentinel” cancer allowing identification of patients with mutations in MMR genes [98].

Investigations of screening strategies mainly focused on the most frequent hereditary syndrome, represented by the LS. Due to their low prevalence, Cowden and other hereditary tumor syndromes with EC risk are just rarely diagnosed in an unscreened EC population. A targeted genomic search for specific non-LS mutations appears to be only useful, if individual or family history suggests so. With the Amsterdam criteria and the Bethesda guidelines, two criteria sets mainly based on patients history have been established to identify LS individuals and screen individuals at risk for further genetic testing [99].

Several cohort studies have investigated the best screening strategy and target group for LS associated EC [39,100,101,102,103,104,105] (Table 2, see Figure 2 for PubMed search strategy). Despite the lower mean age of patients with hereditary forms of EC, a considerable number of LS-associated cancers (62% and 64% in the two biggest cohorts) were diagnosed after the age of 50, and about 37% of cases did not fulfill clinical criteria (Amsterdam/revised Bethesda) especially in cases age 50 and older. These numbers demonstrate that a screening strategy based on individual or family history or early disease manifestation (prior to age 50) alone has a low sensitivity. Therefore, some authors advocate a universal somatic screening strategy for LS independent of these factors below an age threshold (<60 years or <70 years), which still needs to be defined [39,100,101,104].

Immunohistochemical staining for MMR proteins (MLH1, MSH2, MSH6 and PMS2) in tumor specimen can serve as a screening test and is highly concordant with or even outperforms microsatellite instability testing in endometrial cancer [53]. If IHC screening reveals a loss of expression of one or more MMR proteins and if gene methylation can be excluded (*MLH1* promoter), tumor sequencing may uncover the underlying pathogenic variants and germ line testing will reveal whether there is a hereditary predisposition in LS patients [39,100,101,102,104,105,106,107,108,109,110]. MMR protein loss or high microsatellite instability is present in 23–35% of unselected endometrial cancers.

The identification of a pathogenic germline mutation in an endometrial cancer patient (index patient) allows the predictive genetic testing of unaffected family members (individuals at risk). This helps to identify LS among endometrial cancer patients and to counsel and to subject them to intensified cancer screening programs. Importantly, if individual or family history is suspicious and loss of MMR proteins is demonstrated by IHC of tumor tissue, LS must be suspected even if germline analysis cannot confirm the mutation. According to the largest published series, the mutation detection rate (positive predictive value, PPV) of an immunohistochemistry staining of MMR proteins in order to identify LS associated endometrial cancers is 46% if all unselected endometrial cancer cases below age 60 are analyzed, IHC is eventful and methylation as a cause of MLH1/PMS2 loss can be excluded. The authors conclude that this screening strategy provides the highest PPV regarding the identification of mutation carriers with the lowest number of diagnostic tests and therefore appears to be the most effective strategy analyzed in this series [39]. This relatively low PPV for Lynch syndrome is probably due to unrecognized MMR gene somatic mutations, so the underlying tumor biology could still be driven by mismatch repair deficiency.

In a prospective cohort study different screening strategies for LS including IHC, family history and tumor morphology were investigated and compared regarding test performance criteria. IHC in women aged <60 years had the best performance characteristics, with a sensitivity of 100%, a specificity of 86.1%, a positive predictive value of 58.3% and a negative predictive value of 100%. Family history and tumor morphology both had the lowest sensitivity at 71.4%. Overall tumor morphology had the poorest performance, with a specificity of 42.1% [101].

Hence, IHC for MMR proteins has proven to be most sensitive and may be favored over MSI (microsatellite instability) testing, which still can serve as a backup strategy. Patients with MMR-deficient tumors could then undergo targeted next-generation-sequencing and methylation testing of their tumor tissue and/or, if family history or age criteria are suspicious, targeted next-generation sequencing of MMR genes in their germline (blood) samples. Possible paths to LS diagnosis are illustrated in Figure 3.

### 2.2. Gynecologic Surveillance in Families with Hereditary EC

Recommendations for gynecologic surveillance strategies in LS families differ between guidelines and countries. Overall, there is no evidence to support a systematic screening for endometrial cancer as no study has demonstrated a true survival benefit. In an unscreened population, endometrial cancers are mostly detected at early stages as most postmenopausal patients present with uterine bleedings and seek medical advice early. Gynecologic examination and finally hysteroscopy, dilation and curettage will lead to the diagnosis. At early stages, standard therapy consisting of total hysterectomy plus bilateral salpingo-oophorectomy (BSO) and pelvic washings will result in a remarkable 5-year survival rate between 75% and 83%. Considering the overall good prognosis of EC due to usually early detection, it appears difficult to demonstrate a true survival benefit of a general systematic screening strategy even in a high risk group.

However, as LS patients on average are diagnosed twenty years earlier than patients with sporadic endometrial cancer, they are not rarely pre- or peri-menopausal and irregular bleedings as a cardinal symptom of endometrial cancer harder to identify in this group. Therefore EC might potentially be detected later and at later stages. This provides the rationale for a systematic screening especially in the population of pre- and perimenopausal LS mutation carriers. A number of cohort studies investigated which screening strategy is the most effective (Table 3, see Figure 2 for PubMed search strategy). Transvaginal ultrasound alone seemed not to be sufficient for early detection of endometrial cancers in LS patients especially in pre- and peri-menopausal patients [111,112,113]. In another study on 175 LS patients (759 patient years) investigating transvaginal sonography (TVS) in combination with endometrial biopsy (EB), 4 of 14 endometrial cancers were detected by TVS and 8 by EB alone [114], which in addition also detected 14 premalignant hyperplasias in the cohort.

The results of three prospective endometrial cancer screening studies are inconsistent. However, these studies had a low number of cases and short follow-up. In 58 LS patients, 2 endometrial cancers were detected combining TVS and EB [115]. A second study on 41 LS patients screened by annual TVS, outpatient hysteroscopy and EB half of all endometrial cancers were not detected by TVS, concluding that EB enhanced sensitivity of screening significantly [116]. These results are contrasted by the results of the third study on 75 LS patients (300 women years) in which all 6 premalignant lesions and endometrial cancers were detected by ultrasound [117].

Taking the published data on screening together, the evidence for a general endometrial cancer screening in LS or CS patients remains weak. However, there is evidence that annual EB is superior to TVS for identification of endometrial cancer and other premalignant endometrial lesions. Although a group of European experts recommends a screening for all mutation carriers using TVS and EB starting from age 35 to 40 [49], the best age of onset of cancer surveillance remains unclear. A retrospective cohort study utilizing the Genetic Register Lynch Syndrome Database from the Manchester Centre for Genomic Medicine identified 568 female LS carriers and investigated whether mutated gene and type of mutation influence age at onset of LS associated cancers [118]. In this cohort 162 endometrial cancers (38%) were diagnosed. Of all cases, 38% were attributable to *MSH6*, 30% to *MSH2* and 27% to *MLH1* mutations. Women with *MSH6* mutations with endometrial cancer were older compared to women carrying other mutations. The mean ages of endometrial cancer onset were 49 (range 17–71), 47 (range 32–72) and 53 (range 42–66) for women with *MLH1*, *MSH2* and *MSH6* mutations, respectively. There were no endometrial cancer patients with *PMS2* mutations in this series. In the Prospective Lynch Syndrome Database (PLSD), as discussed above, the risk for endometrial cancer at age 75 in *PMS2* variant carriers was estimated at only 13% [32]. This would suggest a potential role for a stratified rather than uniform surveillance strategy depending on the underlying genomic alteration.

### 2.3. Prophylactic Surgery

BSO should be considered when child-bearing is completed ideally at around age 40 [119]. A retrospective analysis revealed a significant decrease in endometrial cancer incidence after prophylactic hysterectomy in LS patients. However, prospective data showing long-term quality of life and investigations on potential negative effects are still missing [120]. In view of the high lifetime risk to develop endometrial cancer, also the psychological benefit after completion of family planning and prophylactic hysterectomy needs to be considered. When offered, many LS patients opt for a prophylactic operation. German colon cancer guidelines recommend offering prophylactic hysterectomy if laparotomy or laparoscopy is scheduled for other reasons to avoid re-laparotomy/laparoscopy when endometrial cancer is diagnosed later in life.

Basis for the recommendation of prophylactic BSO is the fact that there are no effective screening strategies for ovarian cancer available as demonstrated in numerous studies on patients with hereditary breast and ovarian cancer. The only effective measure to improve survival is prophylactic BSO. In LS, the risk for ovarian cancer is increased with a lifetime incidence of about 12% but to a lesser extent than in carriers of a pathogenic *BRCA1* or *BRCA2* variant. However, negative effects of premenopausal BSO and therefore endocrine deprivation on quality of life, sexual function, bone density and cardiovascular morbidity also need consideration. Moreover, OC risk for pathologic *MSH6* variants carriers appears to be low and is not even measurable in *PMS2* variant carriers [28]. According to the prospective Lynch syndrome database the prognosis of LS associated OC with an approximate 10 year survival of over 80%, which is significantly better than survival in pathologic *BRCA1* or *BRCA2* variant carriers [28]. Accordingly, a general recommendation regarding prophylactic BSO in particular for premenopausal LS patients cannot be given. However, all patients need counseling regarding the pros and cons in order to make an informed choice for or against prophylactic BSO.

In a series of 39 prophylactic hysterectomy specimens in LS patients with a mean age of 45 years and without clinical or imaging signs of endometrial cancer, Bartosch and coworkers [121] found three invasive endometrial cancers and four atypical hyperplasias, which are considered to be precancerous lesions. This finding of incidental endometrial cancers and atypical hyperplasias was also confirmed in case series of other investigators, demonstrating that prophylactic surgery is effective and can prevent advances stage disease in asymptomatic patients [122].

### 2.4. MSI and Defective Mismatch Repair as Therapeutic Target in Mutation Carriers with EC

There are clinical trials ongoing for metastatic EC investigating the effect of targeted or immunotherapy drugs. POLE mutated ECs and ECs with MSI similar to triple negative breast cancers express multiple neoantigens. These interact with activated T-lymphocytes and thereby trigger a potentially strong antitumor immune response in the tissue. Accordingly, recent studies found LS EC to be highly immunogenic [123,124].

Tumor cells expressing Programmed cell death 1 ligand 1 (PD-L1, also known as CD274) bind the T-cell receptor PD-1, which suppresses the immune response and helps tumor cells to survive. The PD-L1/PD-1 interaction is targeted by check point inhibitors like pembrolizumab. The FDA has recently granted an accelerated approval to pembrolizumab for the treatment of cancers with defective MMR-repair (including EC) based on the positive results of a phase two basket study including various cancer types (Keynote-028) [125]. At the 2019 ESMO conference, the results of a phase Ib/II Study (Keynote-146) investigating the combination of pembrolizumab and lenvatinib (oral multikinase inhibitor targeting VEGF-1–3, FGFR 1–4, PDGFRα, RET and KIT) in 124 patients with progressive metastatic EC after ≤2 courses of chemotherapy were presented. The objective response rate was 38% in the overall cohort, and remarkable 63.6% for patients with MSI or defective MMR [126].

Preclinical studies have further suggested that Poly-ADP-Ribose-Polymerase (PARP) inhibitors, perhaps in combination with PI3-kinase inhibitors, could be useful in endometrial cancer treatment as most endometrial tumors are PTEN-deficient [127,128]. Twelve clinical trials including PARP inhibitors are presently ongoing [129]. Such medication may also prove effective in high-grade non-endometrioid endometrial cancers that have been shown to be deficient in homologous recombinational DNA repair [130].

## 3. Conclusions

Endometrial cancer is a genetically heterogeneous disease with a prominent contribution of mismatch repair gene mutations that cause Lynch Syndrome. Immunohistochemical staining of MMR proteins is an effective screening strategy to identify those patients and to provide genetic counseling and cancer surveillance to their blood relatives. Prophylactic hysterectomy and bilateral salpingo-oophorectomy are effective preventive surgeries but need to be considered on an individual basis. With the identification of frequently targeted cancer driver genes, stratified treatments are emerging and might prove useful in the future. A few endometrial cancer susceptibility genes beyond Lynch Syndrome have been identified more recently, but their pathogenic variants are collectively rare. More common polymorphisms contribute to endometrial cancer risk in an additive fashion and have already provided insight into novel disease-associated molecular pathways. Larger case-control studies are needed to fully explore the genomic landscape of endometrial cancer predisposition.

## Figures and Tables

**Figure 1 cancers-12-02407-f001:**
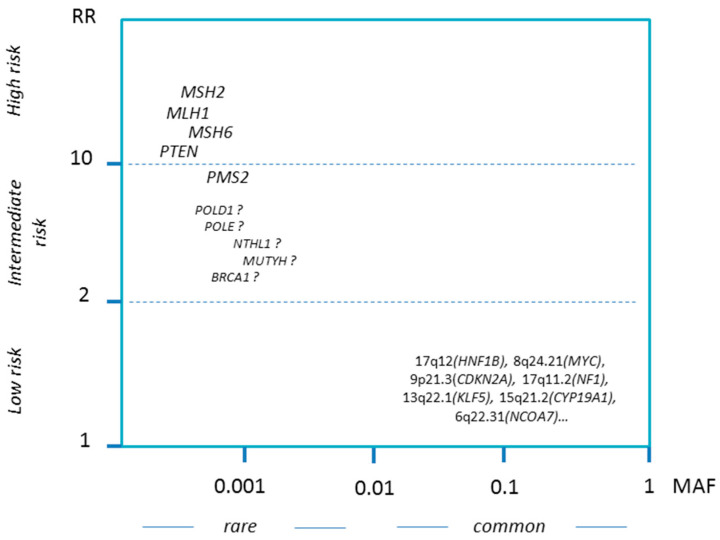
Schematic illustration of hereditary risk factors with risk estimates plotted against frequency. RR, relative risk; MAF, minor allele frequency. Risk estimates refer to classical pathogenic variants and may vary depending on the nature of the specific gene variant. Candidate intermediate risk factors are indicated with question mark (see Section 1.3 for further discussion). Low-risk genome-wide case-control association studies (GWAS) loci are indicated by their chromosomal region with plausibly predicted nearby genes given in brackets. MAF values refer to the abundance of loss-of-function variants estimated from the gnomAD database (https://gnomad.broadinstitute.org/). As a note of caution, these MAFs may substantially differ between distinct populations, some missense variants are not covered and only a subgroup of variants in *POLD1* and *POLE* may cause cancer (see Section 1.3.4). Figure not drawn to scale.

**Figure 2 cancers-12-02407-f002:**
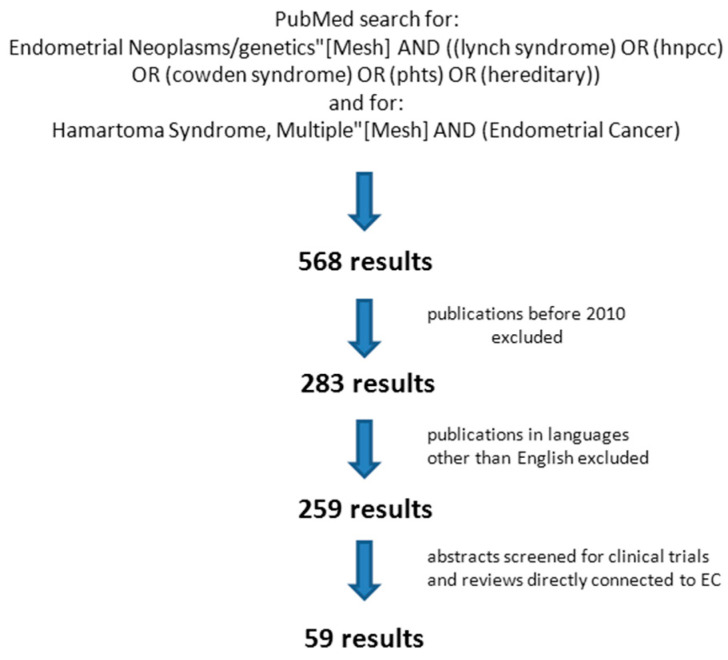
PubMed search strategy.

**Figure 3 cancers-12-02407-f003:**
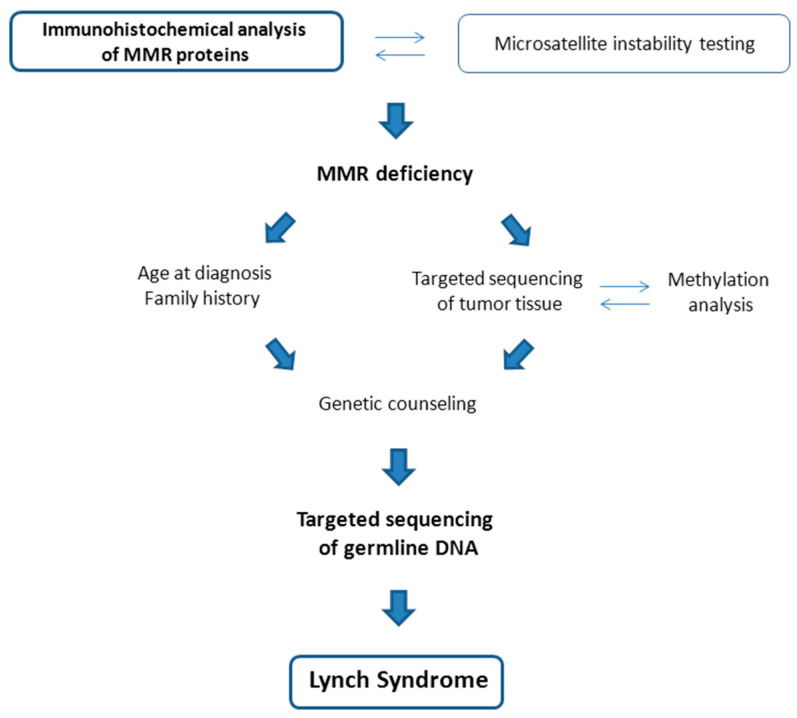
Paths to a Lynch Syndrome diagnosis, starting from the detection of MMR (mismatch repair) deficiency through IHC (immunohistochemical) or MSI (microsatellite instability) analysis. Patients with clinical criteria of Lynch Syndrome may be tested directly for germline predispositions (left arm). Targeted sequencing of MMR genes in tumor tissue, perhaps complemented by methylation analyses, may also reveal pathogenic variants that would qualify to be tested in the germline and may ultimately lead to a LS diagnosis (right arm).

**Table 1 cancers-12-02407-t001:** Established and candidate endometrial cancer risk genes.

Gene	Location	Frequency *	Associated Syndrome		Lifetime Risk EC	Other Cancers
			Biallelic	Monoallelic		
Established						
*MLH1*	3p22.2	1/1500	CMMRD	Lynch Syndrome	~40–45%	Colon, Ovary, Stomach, Pancreas, Brain **, HS **
*MSH2*	2p21-16	1/10,000	CMMRD	Lynch Syndrome	~50%	Colon, Ovary, Skin, Brain **, HS **
*MSH6*	2p16.3	1/2500	CMMRD	Lynch Syndrome	~40–45%	Colon, Ovary, Stomach, Pancreas, Breast?, Brain **, HS **
*PMS2*	7p22.1	1/600	CMMRD	Lynch Syndrome	~15–20%	Colon, Ovary, Breast?, Brain **, HS **
*PTEN*	10q23.31	1/10,000	unknown	Cowden Syndrome	~25%	Breast, Thyroid, Kidney, Colon, Skin
Candidates						
*POLD1*	19q13.33	not defined *	unknown	Lynch Syndrome-like	likely increased	Colon
*POLE*	12q24.33	not defined *	(FILS or IMAGEI Syndrome)	Lynch Syndrome-like	likely increased	Colon
*NTHL1*	16p13.3	1/250	NTHL1 multicancer syndrome	–	likely increased for homozygotes	Colon, Breast
*MUTYH*	1p34.1	1/200	MUTYH-associated polyposis	MUTYH-associated polyposis	possibly increased for homozygotes	Colon
*BRCA1*	17q21.31	1/600	Fanconi Anemia S	Hereditary Breast and Ovarian Cancer	possibly increased for serous EC	Breast, Ovary, Skin, Pancreas

Established and candidate endometrial cancer risk genes. Summarized for each gene are chromosomal location, carrier frequency estimates for loss-of-function variants in the general population, clinical syndromes associated with biallelic or monoallelic variants, estimated lifetime risk for endometrial cancer and common cancer sites apart from the endometrium. Lifetime risks for established genes refer to heterozygote risks. * carrier frequency for loss-of-function variants does not cover non-truncating variants. Note that *POLD1* and *POLE* truncating variants are associated with FILS or IMAGEI syndromes while cancer-associated variants are missense variants. ** Brain tumors and hematological cancers are common in patients with biallelic mismatch repair (MMR) gene variants. ES, endometrial cancer; HS, hematopoietic system; CMMRD, “Constitutional MisMatch Repair Deficiency”; FILS, “Facial dysmorphism, Immunodeficiency, Livedo, and Short stature”; IMAGEI, “Intrauterine growth retardation, Metaphyseal dysplasia, Adrenal hypoplasia congenita, GEnital anomalies, and Immunodeficiency”.

**Table 2 cancers-12-02407-t002:** Cohort studies investigating screening strategies for hereditary syndromes among endometrial cancer (EC) patients.

Study	Number	Design	Patients	Strategy	Results
Backes et al., 2009 [94]	*n* = 140	Prospective	Unselected EC patients	IHC MMR proteins → invitation for genetic counseling when suggestive for LS.	30 patients (21%) with loss of one or more MMR proteins, 15/30 invited to genetic counseling, 2/15 accepted both negative for LS.
Buchanan et al., 2014 [35]	*n* = 702	Prospective (multicentric)	Unselected EC patients	IHC MMR proteins + DNA *MLH1* methylation status for all tumors exhibiting MLH1/PMS2 loss → genetic testing in IHC MRD patients.	170 (24%) of 702 patients showed MMR loss. 158/170 available for genetic testing. 22/158 truncating gene variants. Overall carrier frequency 3%. Testing MMR loss by IHC in women <60 years at diagnosis was optimal regarding sensitivity and cost-effectiveness.
Egoavil et al., 2013 [88]	*n* = 173	Prospective (monocentric)	Unselected EC patients	MMR-IHC and MSI testing MMR mutation testing in positive cases. If MMR gene mutation was detected or *MLH1* methylation in the blood test was positive, patients were classified as LS positive.	61/173 patients had abnormal IHC or MSI results. 8/61 patients tested positive for LS (prevalence 4.6% (8/173)).
Ferguson et al., 2014 [89]	*n* = 117	Prospective (monocentric)	Unselected EC patients	Family history assessment, IHC screening for MMR, MSI testing, tumor morphology followed by germline testing for MMR gene mutations.	34/117 had MMR deficiency in IHC. 27/117 had MSI, 7/27 LS (5.9%). IHC < 60 had sensitivity of 100%, specificity of 86.1%, with PPV of 58.3% and NPV of 100%, family history and tumor morphology had poorest performance with a specificity of 42.1%.
Gausachs et al., 2012 [96]	n.a.	n.a.	*n* = 122 CRC patients with MMRD, 57 LS, 48 MSS cancers and positive family history for CRC, 73 sporadic CRC.	*BRAF* mutation and *MLH1* promoter hypermethylation were assessed and a decision model was developed to estimate incremental costs of alternative case finding methods for detecting *MLH1* mutation carriers.	Sensitivity of the absence of *BRAF* mutations for depiction of LS patients was 96% (23/24) and specificity was 28% (13/47). Specificity of *MLH1* promoter hypermethylation for depiction of sporadic tumors was 66% (31/47) and sensitivity of 96% (23/24). The cost per additional mutation detected by hypermethylation analysis lower when compared with *BRAF* and germinal *MLH1* mutation study.
Hampel et al., 2006 [90]; Hampel et al., 2007 [91]	*n* = 562	Prospective (multicentric)	Unselected EC patients	MSI testing, if positive germline mutations in MMR genes were tested.	119/562 were MSI positive, 11 germline mutations in at least one MMR gene, one patient not MSI positive but germline mutation in *MSH6*, one patient’s MSI test failed. 8/13 patients w/o criteria for HNPCC syndrome, 8/13 diagnosed >50 years.
Leenen et al., 2012 [92]	*n* = 179	Prospective (multicentric)	Unselected EC patients	MSI/IHC for MMR proteins. *MLH1* promoter hypermethylation if MSI high and MLH1 absent. Tumors classified as: (1) likely to be caused by LS, (MSI high and MMR protein deficiency) (2) sporadic MSI-H (MSI high, MLH1 absent, and *MLH1* promoter hyper-methylated), or (3) MSS.	Eleven EC patients found likely to have LS (6%) Germline analyses revealed 7 MMR mutations. Ten patients likely to have LS (92%) were >50 years. 31 sporadic MSI-H tumors with *MLH1* promoter hypermethylation (17%; 95% CI 13–24%) identified.
Moline et al., 2013 [95]	*n* = 245	Prospective	EC patients <50 years or suspicious personal history or histo-pathologic features. EC <69 years or at any age with suspicious features	MSI and IHC, later IHC for two proteins, and *MLH1* promoter methylation analysis when indicated. Genetic counselor contacted patients to offer counseling appointments.	245 EC screened. 62 (25%) abnormal results, 42 patients referred for genetic counseling. 34/42 patients underwent genetic counseling, 28 pursued genetic testing, 11 LS. Age and pathology overlooked 27 eligible cases, 3 cases of LS were only found by clinician request.

Cohort studies investigating screening strategies for hereditary syndromes among EC patients (IHC, Immunohistochemistry; MMR, Mismatch repair; LS, Lynch-Syndrome; MMRD, Mismatch Repair Deficiency; MSI, microsatellite instability; MSS, microsatellite stability; n.a., not applicable; NPV, negative predictive value; PPV, positive predictive value; CRC, colorectal cancer; EC, endometrial cancer).

**Table 3 cancers-12-02407-t003:** Studies on different gynecologic surveillance strategies in families with hereditary EC.

Study	Number	Design	Patients	Intervention	Results
Dove-Edwin et al., 2002 [100]	*n* = 269	Prospective (multicentric)	Unselected women from HNPCC or HNPCC-like families.	Annual or biannual TVU.	During surveillance two EC, none detected by screening.
Helder-Woolderink et al., 2013 [106]	*n* = 75	Prospective (monocentric)	Women >30 years with LS or first-degree relatives with LS.	Period 1 TVU and CA 125; Period 2 TVU, CA 125 and EB to detect EC or precancerous lesions.	Six pre-malignancies and one EC detected. 0/6 would have been missed without EB, annual TVU seems to detect pre-malignancies in women with LS or first-degree relatives with LS.
Lécuru et al., 2008 [104]	*n* = 62	Prospective (monocentric)	Unselected women with LS/meeting Amsterdam II Criteria.	Women with least one hysteroscopy and EB during standard screening.	Three possibly malignant lesions detected, none of them missed w/o hysteroscopy due to abnormal uterine bleeding.
Manchanda et al., 2012 [105]	*n* = 41	Prospective observational	Unselected women with LS.	Annual OHES vs. annual TVS.	OHES detected 4/4 EC/AEH, TVS 2/4; OHES has similar specificity, higher PLR and lower NLR.
Renkonen-Sinisalo et al., 2007 [103]	*n* = 175	Prospective Cohort Study	Unselected women with LS.	TVU and EB.	14/175 patients diagnosed with EC. 11/14 diagnosed by surveillance. 4/11 diagnosed by TVUS only. EB detected 14 cases of potentially premalignant hyperplasia. Cases detected by surveillance at more favorable disease stage. 0/14 detected patients but 6/83 symptomatic LS patients died of EC (n.s., *p* = 0.4).
Rijcken et al., 2003 [101]	*n* = 41	Prospective	Women with LS.	Annual TVU and serum level CA 125.	17/179 TVUs suggested biopsy. 3/17 AEH. One EC as an interval carcinoma, no OC.
Ryan et al., 2017 [107]	*n* = 162	Retrospective	Unselected women with LS diagnosed with EC.	Comparison of mutated MMR genes and type of mutation.	Patients with *MSH6* variants and those with truncating *MLH1* variants diagnosed with EC at later age (median difference 6.6 years; 95% CI 2.7–10.4; *p* = 0.002 for truncating *MLH1* variants).

Cohort studies on different gynecologic surveillance strategies in families with hereditary EC. (HNPCC, hereditary non-polyposis coli; TVU, transvaginal ultrasound; TVS, transvaginal sonography; CA, cancer-antigen; EB, endometrial biopsy; OHES, Outpatient hysteroscopy and endometrial sampling; EC, endometrial cancer; AEH, atypical endometrial hyperplasia; PLR/NLR, positive/negative likelihood ratio; OC, ovarian cancer).

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
