# Peer review of "Genetic Susceptibility to Endometrial Cancer: Risk Factors and Clinical Management"

_cancers, 2020, doi:10.3390/cancers12092407_

Round 1

Reviewer 1 Report

Dörk et al presented a non-systematic review of endometrial cancer genetics, risk factors ad clinical management of familial tumors.

Comments:

Point 1 include a mix of information that should be more structured. The authors should include as independent points (1.1, 1.2...) a brief presentation of histological types, the TCGA molecular classification, the risk factors, etc.

The authors should correlate molecular and histological data, since each molecular subtype is represented by different histologies.

The review of risk factors others than hereditary predisposition, should be more extensive. Do the same risk factor occur in endometrioid and serous carcinomas?. Sporadic endometrial endometrioid carcinomas with MMRd have the same risk factors than, for example, copy number low endometrioid carcinomas. Do hormonal factors or obesity impact on the development of endometrial cancer in Lynch syndrome carriers?.

The second hit in Lynch syndrome endometrial carcinomas (page 3)tend to be LOH of the wild type allele. Promoter hypermetylation is the main cause of sporadic tumors with MMRd.

In page 4, the last paragraph discussing Lynch syndrome should be modified indicating that MLH1 promoter hypermetylation should be analyzed in tumors with MLH1/PMS2 loss of expression. A similar and clear statement in page 6 should substitute “If IHC screening reveals loss of expression of one or more MMR proteins and if gene methylation can be excluded (MLH1/PMS2 promoter), germ line testing will identify LS patients”. Promoter methylation of PMS2 is not a cause of MMRd in sporadic endometrial cancer and should not be included when talking about MLH1 promoter hypermetylation.

The authors should indicate the searching strategies for selecting series in Tables 1 and 2. For example, after a non-comprehensive search on PUBMED other series similar to those presented in Table 1 are found (PMID: 27556954, 2796587, 26523542, 28847642, 25617771).

The authors should include in their review the role of somatic mutations of MMR genes as a cause of MMRd In this sense the authors included the following paragraph: “According to the largest published series, the mutation detection rate (positive predictive value, PPV) of an immunohistochemistry staining of MMR proteins in order to identify LS associated endometrial cancers is 46% if all unselected endometrial cancer cases below age 60 are analyzed, IHC is eventful and methylation as cause of MLH1/PMS2 loss can be excluded”. This relatively low PPV for Lynch syndrome is probably due to unrecognized MMR gene somatic mutations.

The authors should include a figure representing the proposed algorism of detection including all available techniques (IHC, MSI, MLH1 promoter methylation, NGs for the detection of somatic mutations in tumor tissue and NGS for the detection of germline mutations in blood samples).

Reviewer 2 Report

Dörk et al. review the genetic risk factors for endometrial cancer and summarize the studies performed on clinical recommendation for detection and management of hereditary endometrial cancer. The manuscript is very nicely written and informs about the current trends in the field.

Following aspects need to be taken into account:

  • The title of the manuscript does not directly communicate the topic of the review, which is inherited EC. The authors should consider adapting the title to "Genetics of hereditary endometrial cancer:…", or "Inherited endometrial cancer: risk factors…"

  • Authors mention the interesting topic of mutational alterations present in normal endometrial epithelium. As authors also discuss alteration in separate glands in the endometrium and discuss the molecular phenotype of microsatellite instability, it would be important to mention the study by McKenzie et al. in Pathology (2016) demonstrating loss of MMR protein expression in crypts with subtle histological changes in a Lynch syndrome patient. This study might also have an implication for early detection of cancer in high-risk patients discussed later by the authors.

  • In Figure 1 authors plot the risk of EC for different EC-associated genes against their mutation frequency in the population. Authors show PMS2 to be relatively rare, but associated with substantial relative risk. Recent studies demonstrated that PMS2 is much more frequent in the population (1:700) compared to MLH1 and MSH2 mutations (Win et al. 2017). Authors need to adapt this figure to better reflect the actual findings of the so far performed studies. In addition, the scale could be complemented and include 0.1 and 0.001 values on the x axis and "zero" should be eliminated.

  • Authors nicely summarize genetic syndromes related to EC risk in the text. It would be very suited for such a review article to have a table summarizing syndrome, responsible genes, their prevalence and penetrance (biallelic/monoallelic separately whenever suitable), cancer in other organs related to this genetic predisposition. Information in such a condensed presentation form would increase the value of the manuscript for clinicians and easily summarize more information at one glance and help to shorten the text.

  • Concerning other tables presented in the manuscript: both tables would benefit from shortening the content of the Results columns to an absolute minimum, as the table should not repeat the text and should be as concise as possible (using only keywords, avoiding re-describing the cohorts).

  • On page 6, section 2.1, first paragraph, authors refer to EC as sentinel cancer in Lynch syndrome. As no citation has been used for this statement, authors can include Lu et al 2005 Obstetrics and Gynecology.

  • On page 9, section 2.2., first paragraph, last sentence authors seem to argue against a systematic EC screening in LS carriers without specifying the population, whereas in the second paragraph they argue for the screening in younger patients. Authors should specify, which population they focus on, when arguing for or against screening.

  • Page 10, second paragraph: authors refer to a screening recommendation via TVS and EB from the age of 35-40, without specifying, which genetic disease is discussed. The cited paper (Vasen et al.) focuses on CMMRD and discusses screening from the age of 20. The syndrome discussed here should be specified and the referenced literature should be checked and complemented, if necessary. Specification of the discussed syndrome applies also to other parts of the manuscript, particularly, it should be made clear what syndrome authors refer to when discussing recommendations.

  • On page 11, section 2.3. authors discuss the clinical value of prophylactic surgery, including bilateral salpingooophorectomy (BSO) and mention the adverse effects of BSO. Additional point that needs to be weighed during decision making is the excellent survival of patients with ovarian cancer in Lynch syndrome, as opposed to BRCA-related ovarian cancer (see PLSD studies, Moller 2017 Gut). This point needs to be taken into account when discussing the clinical recommendation of BSO in patients with such an excellent survival after cancer. Moreover, also here, stratification by gene is very important and needs to be mentioned, as for example, in PMS2 carriers the role of such prophylactic surgery is yet unclear (ten Broeke et al. JCO 2018).

  • On page 11, section 2.4 authors discuss the ongoing clinical trials for treatment of inherited EC, focusing on immunotherapy-based approaches. Authors may want to refer to the studies reporting high immune infiltration in Lynch syndrome EC, one of the pre-requisites for response to immune checkpoint therapy (Ramchander et al. 2020 Frontiers Immunology, Bohaumilitzky et al. 2020 Journal of Clinical Medicine).

Minor points:

  • Page 4, section 1.2 Cowden Syndrome, 3rd sentence: "…a cancer syndrome that affects… and carries an increased lifetime risk…". As the individuals carry risks, I would suggest to reformulate it, for example: "a cancer syndrome that affects… and predisposes to an increased lifetime risk…".

Reviewer 3 Report

The authors have written a comprehensive review about endometrial cancer predisposition and its relevance for clinical management of these patients. The following points should be addressed:

  • The gene POLE/POLD1 should be added to figure one, in spite of the uncertainties concerning the relative risk (which also exist for other genes indicated there with a question mark).
  • In the third paragraph of section 1.1 it is indicated that the second hit commonly occurs via promoter methylation. This is a relatively common mechanism for MLH1, but not for the other mismatch repair genes MSH2, MSH6 and PMS2. Deletion of the second allele and copy number neutral loss of heterozygosity are also common for all four genes, so that sentence needs to be rephrased.
  • In the same paragraph, the authors state that” the hypermethylation can be induced by constitutional deletions of neighboring genes EPCAM (adjacent to MSH2) or LRRFIP2 (upstream of MLH1); these germline deletions thus represent additional hereditary factors”. However, the two situations are very different. Whereas EPCAM deletion are a well-known mechanism of secondary MSH2 hypermethylation and silencing, LRRFIP2 deletions per se are not known to cause MLH1 hypermethyliation. Instead, the few reports that exist show deletions or inversions that affect both LRRFIP2 and MLH1 that either create expressed, but non-functional, gene fusions or inactivation of both genes (Genet Med 2011, 13:895-902; J Med Genet 2011, 48:513-9; Clin Cancer Res 2009, 15:762-9).
  • Primary (that is, not associated with an underlying genetic alteration) constitutional MLH1 hypermethylation as a rare cause of non-transmissible Lynch syndrome is not mentioned in the manuscript.
  • In the 4th paragraph of section 2.1 the phrase “If IHC screening reveals loss of expression of one or more MMR proteins and if gene methylation can be excluded (MLH1/PMS2 promoter), germ line testing will identify LS patients” should be clarified. Promoter methylation of MLH1 but not PMS2 is a common cause of MLH1 and PMS2 loss of expression. Furthermore, not all cases with loss of expression of MMR proteins and absence of methylation will show a germline deleterious variant in the MMR genes (for instance, it could be a sporadic carcinoma with biallelic somatic mutations).

Round 2

Reviewer 1 Report

The authors have answered the comments of this reviewer.

Author Response

Thank you for your time and helpful suggestions.

Reviewer 2 Report

Authors have thoroughy revised the manuscript and implemented the suggestions.

My only additional suggestions would be for the newly prepared table (Table 1):

1. to take "Turcot Syndrome" and "Muir-Torre Syndrome" out, as those are purely clinical definitions;

2. to specify "EC risk" with regard to biallelic or monoallelic mutation carriers, as those values are not the same for these two groups of carriers and this should be made clear for the reader.

Author Response

Thank you for your comments. We have removed the clinical names Turcot Syndrome and Muir-Torre Syndrome from the Table as requested. We cannot reliably specify the EC risk for CMMRD patients since there are too little data available in the literature and several patients die early from other tumors. However, we have added a footnote that the relative risks provided in our table refer to heterozygote risks only.